# SegLLM: Multi-round Reasoning Segmentation with Large Language Models

**XuDong Wang**[*1]    **Shaolun Zhang**[*1]    **Shufan Li**[*2]    **Konstantinos Kallidromitis**[3]
**Kehan Li**[1,4]    **Yusuke Kato**[3]    **Kazuki Kozuka**[3]    **Trevor Darrell**[1]

[1]UC Berkeley    [2]UCLA    [3]Panasonic AI Research    [4]Stanford

## Abstract

We present SegLLM, a novel multi-round interactive reasoning segmentation model that enhances LLM-based segmentation by exploiting conversational memory of both visual and textual outputs. By leveraging a mask-aware multimodal LLM, SegLLM re-integrates previous segmentation results into its input stream, enabling it to reason about complex user intentions and segment objects in relation to previously identified entities, including positional, interactional, and hierarchical relationships, across multiple interactions. This capability allows SegLLM to respond to visual and text queries in a chat-like manner. Evaluated on the newly curated MRSeg benchmark, SegLLM outperforms existing methods in multi-round interactive reasoning segmentation by over 20%. Additionally, we observed that training on multi-round reasoning segmentation data enhances performance on standard single-round referring segmentation and localization tasks, resulting in a 5.5% increase in cIoU for referring expression segmentation and a 4.5% improvement in Acc@0.5 for referring expression localization.

## 1 Introduction

Image segmentation plays a crucial role in numerous computer vision tasks, while traditional methods have been limited to providing segmentation results for close-set categories (Cheng et al., 2022; He et al., 2017) or simple text queries (Ding et al., 2023; Wang et al., 2024b) using CLIP (Ding et al., 2023; Radford et al., 2021) or BERT (Wang et al., 2024b; Devlin et al., 2018) text embeddings as classifiers. Recent advancements in Large Vision-Language Models (LVMs) (Pi et al., 2023a; Zhang et al., 2023a; Lai et al., 2024; Wu et al., 2024; Liu et al., 2024; Touvron et al., 2023; Alayrac et al., 2022; Awadalla et al., 2023; Dai et al., 2024) have reformulated image segmentation as a next token prediction task, enabling segmentation models to engage in natural language conversations with users and reason about the presence, location, and relationships of objects in complex visual scenes. For instance, LISA (Lai et al., 2024), a Language Instructed Segmentation Assistant, produces segmentation masks by incorporating a [SEG] token into its vocabulary, which, when generated, is decoded into the corresponding segmentation mask.

These LLM segmentation models (Lai et al., 2024; Wu et al., 2024; Pi et al., 2023a; Zhang et al., 2023a) typically achieve their localization capabilities by incorporating a decoder that converts the output [SEG] tokens of LLMs into localization results. They are trained on numerous visual queries such as "please find the heart healthy food in the image", where responses include both text outputs and segmentation masks. Essentially, these models are advanced versions of early open-vocabulary segmentation models, with their text encoders upgraded from smaller language models, such as BERT (Devlin et al., 2018), to smarter LLMs, such as Llama (Touvron et al., 2023). Consequently, LLM segmentation models are often evaluated on traditional referring expression segmentation (RES) datasets, such as RefCOCO, which provide a single text query corresponding to each mask. These single-round referring expression segmentation (RES) datasets overlook one of the most remarkable properties of LLMs (Achiam et al., 2023; Team et al., 2023; Touvron et al., 2023; Jiang et al., 2023): generating multi-round responses in a conversational manner. In this paper, we intend to answer the question: *can segmentation models reason about previously segmented objects and conversations, responding to multiple visual and text queries in a chat-like manner?*

Current LLM segmentation or detection models (Lai et al., 2024; Zhang et al., 2023a; Wu et al., 2024), despite their impressive single-round performance, fall short as multi-modal conversation

---

[*]Equal Contribution

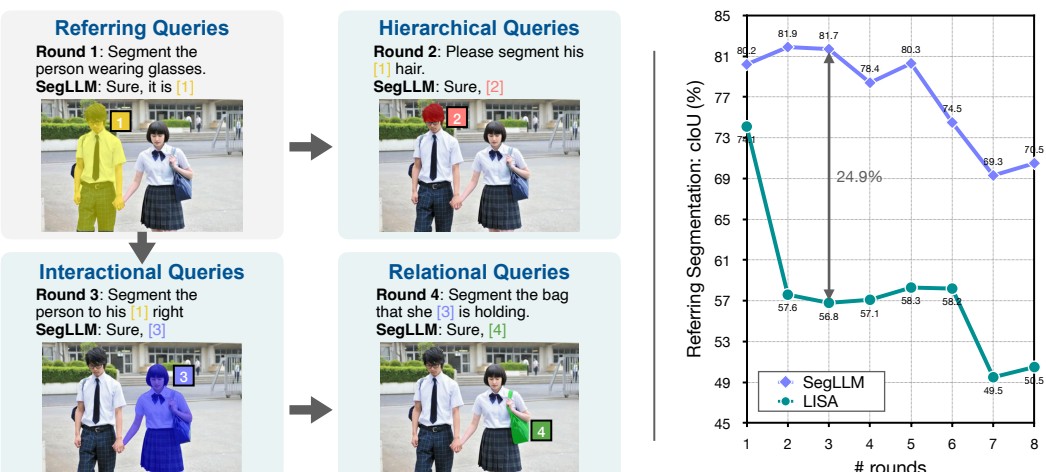

**Figure 1: We present SegLLM, a multi-round interactive reasoning segmentation model** designed to engage in chat-like interactions by responding to both visual and text queries. It reasons about previously segmented objects and conversations to understand complex user intentions. **On the left**: SegLLM can infer intricate relationships between objects, such as positional, interactional, and hierarchical connections with previously identified entities, *e.g.*, instance [1]. **On the right**: We introduce the MRSeg, a new multi-round image referring segmentation benchmark. As the rounds progress, the complexity of interaction and memory retention increases. However, SegLLM consistently surpasses the previous SoTA method LISA (Lai et al., 2024), with a significant margin across all conversational rounds.

agents due to their inability to handle multi-round, interactive conversations. For instance, after obtaining a mask of a *'person in black hoodie'* in Fig. 1, a user might want to perform additional queries based on this mask output—such as segmenting the *'ski he is holding'*, segmenting the *'man standing to the right of him'*, or segmenting a different person if the output is incorrect. Existing models struggle with these complex queries because there is no "communication" between the large language models (LLMs) and the vision encoders. Information flows only from the LLMs to the mask decoder, not vice versa, preventing the LLM from being aware of the output mask and making it difficult to reason about complex queries involving previous mask outputs.

To address this issue, we propose **SegLLM**. Unlike existing LLM segmentation models that naively assemble a mask decoder with an LLM, we introduce a novel communication protocol that feeds the segmentation outputs of the mask decoder back into the input stream of the LLMs, and the past conversation context into the input query of the mask decoder. This design allows the LLMs to "see" past mask outputs and the mask decoder to "see" the past conversation context, enabling it to handle complex queries like *'segment the helmet of the previously segmented person'*, as shown in Fig. 1. Concretely, we introduce a Mask-Encoding scheme to make the LLM mask-aware and a Reference Mask-Decoding scheme to make the segmentation head context-aware. To fully explore the capabilities of these novel designs, we curated multiple high-quality multi-round interactive segmentation datasets, named **MRSeg**. The new dataset consists of complex object queries involving existing mask outputs, formulated in seamless multi-round natural language conversations.

Through extensive experiments, we demonstrate that SegLLM outperforms previous state-of-the-art models by 18∼30% on our multi-round reasoning segmentation benchmarks, MRSeg. Additionally, SegLLM surpasses prior state-of-the-art performance on the single-round referring segmentation and detection benchmark, RefCOCO, with over a 5.5% improvement in segmentation (cIoU) and a 4.5% increase in detection accuracy (Acc@0.5). SegLLM also exhibits greater robustness to various question templates, achieving 9.6% performance gains on RefCOCO with diverse query formats.

# 2 RELATED WORKS

## 2.1 MULTI-MODAL LARGE LANGUAGE MODELS

To leverage the advancements in language models (Brown et al., 2020; Touvron et al., 2023; Chowdhery et al., 2023; Le Scao et al., 2023; Hoffmann et al., 2022) across various modalities, Multi-modal

Large Language Models (MLLMs) have been developed to combine language and vision (Yin et al., 2023; Liu et al., 2024; Zhu et al., 2023; Alayrac et al., 2022). Flamingo was one of the first unified architectures to align image and text pairs in context learning through gated cross-attention blocks (Alayrac et al., 2022). End-to-end MLLMs typically require a finetuning process where an intermediate network (Lai et al., 2024; Zhang et al., 2023a) and/or sampler module (You et al., 2023) is used to map the vision features into the language space. BLIP-2 bridges the modality gap with a querying transformer and a two-stage training process, which involves pretraining on a trainable LLM and instruction tuning on a frozen one (Li et al., 2023b). Models like MiniGPT-4 (Zhu et al., 2023) and LLava (Liu et al., 2024) follow a similar training paradigm, with Vicuna 18 as a language decoder and GPT-4 designed prompts. Other notable models in instruction tuning include Otter (Li et al., 2023a) that is based on (Awadalla et al., 2023), mPLUG-Owl (Ye et al., 2023) with a novel modular architecture, and InstructBLIP (Dai et al., 2024) which features an instruction aware Q-former.

## 2.2 MULTI-ROUND CONVERSATIONAL MLLMS

Recent advancements in MLLMs have focused on enhancing interactive capabilities. Models like Kosmos-2 (Peng et al., 2023) and Shikra (Chen et al., 2023) use visual grounding and referring to provide the LLM with detailed location information of the objects, which enables the user to point out specific areas in the image. Various works aim to improve local information, such as Ferret (You et al., 2023) and PerceptionGPT (Pi et al., 2023b) which employ flexible continuous representations to handle different shapes. Other approaches (Yang et al., 2023a;b; Zeng et al., 2022) utilize prompt engineering and APIs to facilitate interaction, instead of relying on end-to-end models.

More recent approaches introduce the concept of reasoning, leveraging LLMs to provide a visual answer based on implied information. DetGPT (Pi et al., 2023a) performs object detection using high-level instructions rather than distinct classes. GPT4RoI (Zhang et al., 2023b) receives spatial boxes as input to focus on specific regions and better align vision and text. LISA (Lai et al., 2024) adds a new embedding prompt to the mask decoder of the SAM (Kirillov et al., 2023) guiding segmentation, which is then processed by LLaVA (Liu et al., 2024) to perform high-level reasoning. NExT-Chat (Zhang et al., 2023a) expands on LISA by using embeddings instead of tokens for location information and adding a decoder with a joint loss to facilitate object detection.

While some methods support multi-round conversations, they often lack mechanisms to maintain localization performance over successive rounds, leading to degradation and information loss. SegLLM improves the multi-round interactive segmentation by leveraging the text and segmentation results from previous rounds, thereby generating refined masks and supporting hierarchical representations to enhance performance in multi-round interactions.

## 3 BACKGROUND: REASONING SEGMENTATION

**Task definition**. The reasoning segmentation task (Lai et al., 2024) involves generating binary segmentation masks based on an image and descriptive, free-form text prompts. This task requires the model to possess cross-modality comprehension, understanding both the complex visual scenes, as well as the natural-language signals in the text prompt. Specifically, the model must interpret complex user text prompts that go beyond simple class names to include implicit descriptions that require general world knowledge, such as "the device that can illuminate a dark room".

**Overall pipeline**. To achieve such capabilities, reasoning segmentation model typically first employs a pre-trained large multimodal models (VLMs), $\mathcal{F}_{\text{MM}}$, which is capable of comprehending both visual and textual information simultaneously (Lai et al., 2024). A new [SEG] token is then added to the VLMs's vocabulary. Given an input image $x_{\text{img}}$ and input text prompt $x_{\text{txt}}$, the VLMs generates an output text response $\hat{y}_{\text{txt}}$, which includes the [SEG] token to request the generation for a segmentation mask. Finally, the segmentor $\mathcal{F}_{\text{SEG}}$ uses the last layer's hidden state, $h_{\text{seg}}$, corresponding to the [SEG] token along with the input image $x_{\text{img}}$ to generate the segmentation mask $\hat{y}_{\text{SEG}}$.

**Model architecture**. An image reasoning segmentation model, $\mathcal{F}_{\text{MM}}$, typically consists of three key components (Lai et al., 2024): an image encoder $\mathcal{E}_{\text{MM}}$ (*e.g.*, CLIP (Radford et al., 2021) and DINOv2 (Oquab et al., 2023)), a base language model $\mathcal{L}$ (*e.g.*, Llama (Touvron et al., 2023)), and a vision-to-language projection layer $f_{\text{VtoL}}$, which is typically an MLP layer. Given a pair of input image and text prompt $(x_{\text{img}}, x_{\text{txt}})$, the image encoder first encodes the input image into patch embeddings

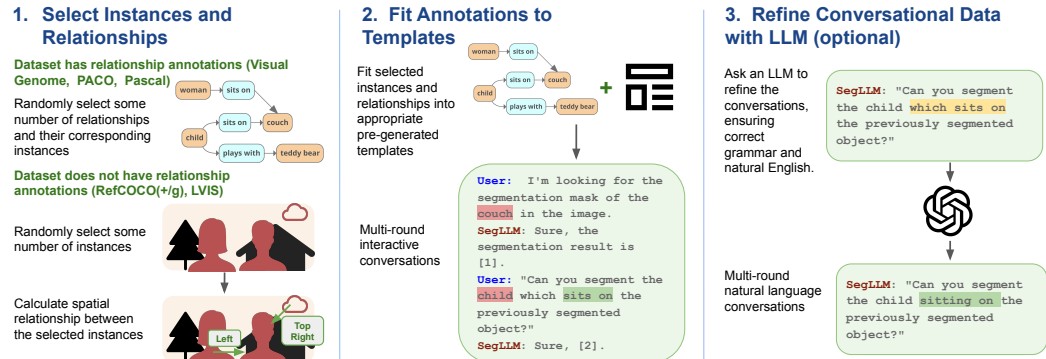

**Figure 2: Pipeline for generating our multi-round conversational dataset MRSeg.** The workflow involves selecting instances, generating relationships, fitting the instances and relationships into conversational templates, and refining the conversations using a language model for improved accuracy.

$h_{\text{img}}$, which are then projected into the text embedding space via $f_{\text{VtoL}}$. The resulting visual tokens are concatenated with the sequence of text tokens $h_{\text{txt}}$. Finally, taking both visual and language tokens as inputs, the language model $\mathcal{L}$ produces the output response $\hat{y}_{\text{txt}}$ containing the [SEG] token: $\hat{y}_{\text{txt}} = \mathcal{F}_{\text{MM}}(x_{\text{img}}, x_{\text{txt}}) = \mathcal{L}(\text{cat}([f_{\text{V2L}}(\mathcal{E}_{\text{MM}}(x_{\text{img}})), h_{\text{txt}}]))$. The [SEG] token in the output responses is then decoded into the segmentation mask using the mask decoder $\mathcal{F}_{\text{SEG}}$ of a pre-trained segmentation model, SAM (Kirillov et al., 2023): $\hat{y}_{\text{SEG}} = \mathcal{F}_{\text{SEG}}(x_{\text{img}}, h_{\text{SEG}}) = \mathcal{D}_{\text{SEG}}(\mathcal{E}_{\text{SEG}}(x_{\text{img}}), h_{\text{SEG}})$.

# 4 MULTI-ROUND REASONING SEGMENTATION

The success of our **SegLLM** method relies on two essential components: a comprehensive dataset **MRSeg** that has an extensive collection of **M**ulti-**R**ound interactive **Seg**mentation instructions, and a mask-aware VLMs specifically designed to reason about the conversational history, with a particular focus on the segmentation masks generated in previous interactions.

## 4.1 DATA PIPELINE

**Data sources**. We constructed our multi-round image reasoning segmentation dataset (MRSeg) based on several widely utilized datasets, and include data from the following sources: RefCOCO(+/g) (Yu et al., 2016; Kazemzadeh et al., 2014), Visual Genome (Krishna et al., 2017), PACO-LVIS (Ramanathan et al., 2023), LVIS (Gupta et al., 2019), Pascal Panoptic Part (de Geus et al., 2021), ADE20K(Zhou et al., 2017), COCO-Stuff(Caesar et al., 2016) and MSCOCO(Lin et al., 2014b). We used bounding box or segmentation annotations from these datasets to generate natural language conversations, applying a template-based approach as detailed in subsequent sections. The overall pipeline can be seen in Fig. 2 and we provide the statistics and some sample data for MRSeg in Fig. 3.

**Multi-round conversation generation**. We design various pipelines for generating multi-round conversations, tailored to the types of data and inter-instance relationships they support:

- **Hierarchical Relationships** (PACO-LVIS, Pascal Panoptic Part): In these queries, the model is tasked with segmenting objects that are sub-parts of previously segmented instances. The queries start by asking about the instance, followed by questions about its parts. Example query: *"Can you segment the <part> of the <object>?"*

- **Positional Relationships** (RefCOCO(+/g), LVIS): These queries require the model to segment objects based on their positional relationships to previous outputs. An example query is: *"Can you segment the <class> that is <relationship> the output from round ?"* We refine these conversations using GPT-4 (our full prompt to GPT-4 can be found in Table A3) to ensure natural language fluency. Details on the RefCOCO(+/g) pipeline are in Fig. A2. Additionally, we introduce a challenging variant called MRSeg (hard), where understanding previous round information is necessary to correctly segment the current instance (details in Appendix A.1).

- **Interactional Relationships** (Visual Genome): Utilizing Visual Genome (VG) relationship annotations, we construct conversations that focus on interactional dynamics, rather than just

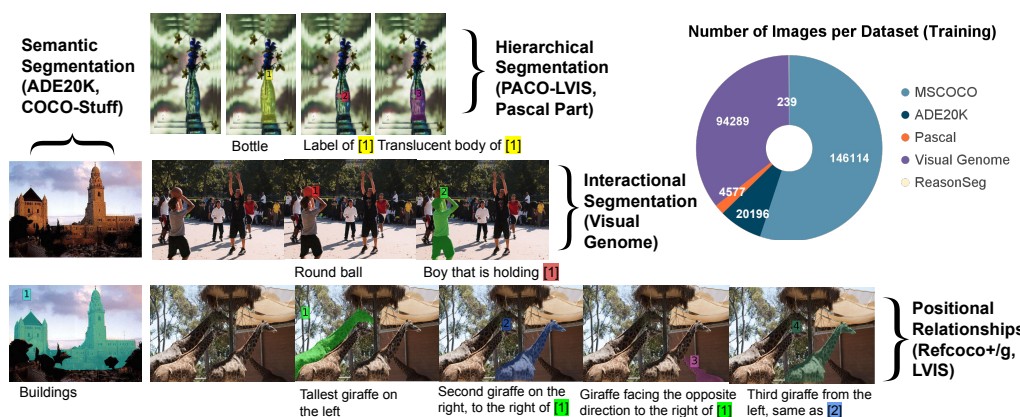

**Figure 3: Statistics and sample conversations for the Multi-Round Referring Segmentation dataset** (MRSeg). We provide more details for MRSeg in Appendix A.1.

positional relationships. Each conversation has two rounds: the first round segments the subject, and the second round segments an object based on its relationship to the subject.

- **Attribute-oriented Queries** (MSCOCO): These queries ask the model to segment objects based on their attributes or usage rather than class names. An example query is: *Q: Outline and extract the object that has a tall, slender neck covered with a distinct pattern of patches. A: Yes, the figure you specified for segmentation is a giraffe*. We generate captions by cropping MSCOCO instances and using GPT-4V prompts (details in Table A2).
- **Single-Round Semantic Segmentation** is based on ADE20K and COCO-Stuff datasets. We construct single-round conversations by fitting class labels into various query templates.

Additional details on the multi-round data pre-processing for MRSeg are provided in Appendix A.1.

**Conversation templates**. We observed that current state-of-the-art chat-based image segmentation models, such as LISA (Lai et al., 2024), tend to rely heavily on a fixed set of question templates. This leads to fluctuations and instability in segmentation quality when user prompts are phrased differently, suggesting potential overfitting to specific language prompts. To address this, we leveraged the web-version of GPT-4 (Achiam et al., 2023) to generate diverse templates, creating more natural language conversations from dataset annotations. We generated templates for direct referring segmentation queries, relational queries, and hierarchical queries. For each query type, we created 100~200 templates for training and 50~100 different templates for validation.

## 4.2 SegLLM for Multi-round Image Reasoning Segmentation

**Overall Pipeline**. We introduce SegLLM to ensure that the VLMs's next token predictions can incorporate the conversational memory from previous interactions, including the visual outputs, *i.e.*, segmented masks, and the text conversations. The architecture of our model is illustrated in Fig. 4. SegLLM consists of two key components: 1) Mask-Encoding Module: This module feeds the output masks back into the input stream of the LLM, enabling it to reason about segmented masks from previous rounds. 2) Mask-Aware Decoding Module: This module allows the mask decoder to generate new masks based on both the visual and textual conversational history, enhancing its contextual understanding. For example, when a user requests segmentation of a part of an object identified in a previous round (*e.g.*, the ear of a man), the model's ability to access prior mask data enables the decoder to more precisely localize and segment the specified object.

**Mask-Encoding Scheme**. For each mask generated by the decoder, we compute two types of embeddings: mask embedding and bounding box embedding. The mask embedding captures the semantic information of the masked object, and the bounding box embedding captures its location within the original image. Employing mask embeddings (one token per mask) and box tokens instead of a new set of patch tokens for each mask substantially reduces the number of visual tokens required for multi-round conversations. Furthermore, since the visual patch tokens for the entire image have already been utilized in previous conversations, this design does not compromise the richness of information necessary for accurate image segmentation and visual question answering.

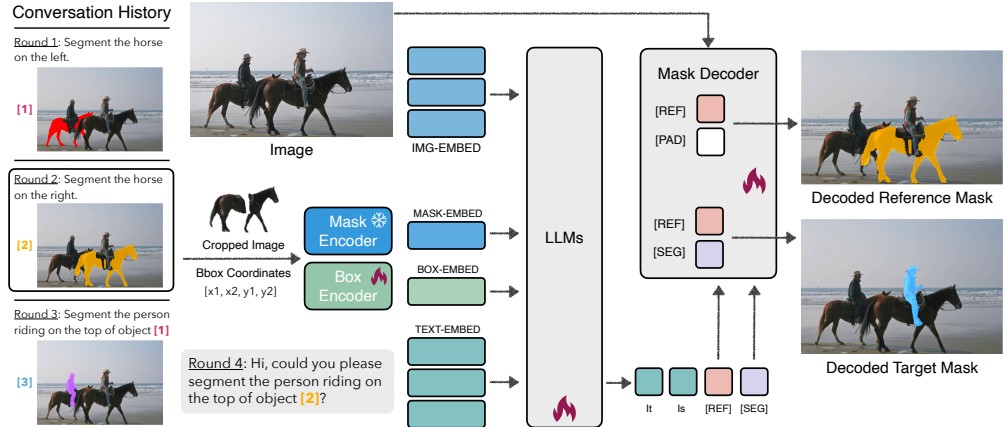

**Figure 4: Model architecture of SegLLM** for multi-round interactive image reasoning segmentation, which can understand complex user intentions and segment entities based on their relationships with previously identified ones. To facilitate this, first, we implement a mask encoding scheme that reincorporates the reference mask information back into the input stream of the LLMs. This enables the LLMs to reason about segmented masks from previous rounds. Second, we develop a mask-aware decoding scheme that allows the mask decoder to generate new masks based on both the output from the LLMs and the historical memory of output masks. The model uses the last layer hidden states associated with the [REF] and [SEG] tokens to generate both the reference mask and the target mask, seamlessly integrating past and current segmentation results.

To obtain the **mask embedding**, we first set the pixels outside the reference mask as black, then we crop the image according to the bounding box of the reference mask. This yields an object-centric image of the masked object. We pass this image to a CLIP encoder (Radford et al., 2021) to get the raw mask embedding. We use an MLP layer to map this embedding to the input dimension of LLMs.

To obtain the **bounding box embedding**, we first compute the bounding box coordinates using the generated mask, then we create a positional embedding whose dimension matches the input dimension of LLM. We use this generated positional embedding as the final bounding box embedding.

For each mask, we obtain the two embeddings and feed them sequentially back to the input stream of LLMs. Following LISA, we use a [SEG] token to generate the masks. During the training process, we employ the teacher enforcing (Williams & Zipser, 1989) and directly append the ground truth mask and bounding box embedding after each [SEG] token. At the inference time, we compute the two embeddings for each mask generate and insert the embeddings before the input for the next round.

**Mask-Aware Decoding.** To facilitate the decoding process, we generate two tokens and [SEG] to the mask decoder. The [REF] token contains information about the referenced mask and [SEG] token contains information about the relational query. For example, in the query "segment the head of *[instance 1]*" where "*[instance 1]*" is a previously segmented person, the [ref] token should encode the previous mask *[instance 1]* while [SEG] should encode the target mask. In the training process, we construct two queries. We match first query "[REF] , [PAD] " to the referenced mask $M_{\text{ref}}$ (*[instance 1]* in the previous example), and match the second query "[REF] , [SEG] " to the desired mask $M_{\text{tgt}}$ (the head of the person in the previous example). The final loss is formulated as:

$$L_{\text{mask}} = L_{\text{seg}}(F([\text{REF}], [\text{PAD}], M_{\text{ref}})) + L_{\text{seg}}(F([\text{REF}], [\text{SEG}], M_{\text{tgt}})) \qquad (1)$$

where $L_{\text{seg}} = L_{\text{ce}} + \lambda L_{\text{DICE}}$. We apply cross entropy loss and DICE loss to the target mask and reference mask predictions. We set $\lambda$ as 1 by default.

## 5 EXPERIMENTS

### 5.1 IMPLEMENTATION DETAILS

We use a pretrained CLIP-ViT-Large (Radford et al., 2021) with a patch size of 14 as the image encoder, HIPIE-R50 (Wang et al., 2024b) as the mask encoder and LLaVA-v1.5-7B (Liu et al., 2024) as the base language model. Compared with LISA, which has exactly one mask per training sample, SegLLM's setup contains multiple masks per conversation. Hence, we replaced the SAM

| Rounds | MR-RefCOCO | | | | MR-RefCOCO+ | | | | MR-RefCOCOg | | | |
|---|---|---|---|---|---|---|---|---|---|---|---|---|
| | LISA | GLaMM | SegLLM | Δ | LISA | GLaMM | SegLLM | Δ | LISA | GLaMM | SegLLM | Δ |
| # 2 | 60.6 | 59.5 | **81.9** | **+21.3** | 51.4 | 52.9 | **78.0** | **+25.1** | 61.3 | 65.4 | **79.2** | **+13.8** |
| # 3 | 58.9 | 61.6 | **81.7** | **+20.0** | 51.2 | 58.3 | **78.5** | **+20.2** | 52.1 | 57.8 | **76.0** | **+18.2** |
| # 4 | 61.3 | 59.3 | **78.4** | **+17.1** | 49.0 | 54.2 | **74.3** | **+20.1** | 56.0 | 55.4 | **77.1** | **+15.0** |
| # 5 | 61.0 | 62.6 | **80.3** | **+17.6** | 48.5 | 50.5 | **76.5** | **+26.0** | 47.5 | 49.4 | **66.9** | **+14.0** |
| # 6 | 60.7 | 62.6 | **74.5** | **+11.9** | 45.6 | 54.8 | **73.4** | **+18.6** | 39.9 | 40.8 | **68.9** | **+24.8** |
| # 7 | 54.4 | 52.1 | **69.3** | **+14.9** | 42.8 | 48.4 | **64.0** | **+15.6** | 55.1 | 57.8 | **71.0** | **+13.3** |
| # 8 | 51.9 | 50.7 | **70.5** | **+18.7** | 36.9 | 43.6 | **59.0** | **+15.4** | 36.3 | 38.4 | **54.9** | **+16.5** |

Table 1: **Multi-round referring segmentation** on the proposed multi-round RefCOCO/+/g benchmarks. As the rounds progress, it becomes harder to interact and retain all relevant information, causing the performance measured in cIoU to drop. SegLLM can consistently outperform LISA (Lai et al., 2024) and GLaMM (Rasheed et al., 2024), across a series of rounds by a significant margin on the MR-RefCOCO/+/g benchmarks.

| Methods | RefCOCO | | | RefCOCO+ | | | RefCOCOg | |
|---|---|---|---|---|---|---|---|---|
| | val | testA | testB | val | testA | testB | val | test |
| VLT (Ding et al., 2021) | 67.5 | 70.5 | 65.2 | 56.3 | 61.0 | 50.1 | 55.0 | 57.7 |
| LAVT (Yang et al., 2022) | 72.7 | 75.8 | 68.8 | 62.1 | 68.4 | 55.1 | 61.2 | 62.1 |
| LISA-7B (Lai et al., 2024) | 74.1 | 76.5 | 71.1 | 62.4 | 67.4 | 56.5 | 66.4 | 68.5 |
| NExT-Chat (Zhang et al., 2024) | 74.7 | 78.9 | 69.5 | 65.1 | 71.9 | 56.7 | 67.0 | 67.0 |
| **SegLLM (ours)** | **80.2** | **81.5** | **75.4** | **70.3** | **73.0** | **62.5** | **72.6** | **73.6** |

Table 2: **Comparison between SegLLM and baseline methods on referring segmentation**. Although not specifically designed for single-round referring segmentation, the diverse and challenging multi-round referring segmentation tasks and training data enable SegLLM significantly outperforms previous state-of-the-art methods on standard referring segmentation tasks by a substantial margin. We use cIoU as the main evaluation metric.

ViT-H mask decoder (Kirillov et al., 2023) with a smaller HIPIE-R50 (Wang et al., 2024b) to reduce the computation overhead during the training, We then fine-tune the LLM model and the projector weights $f_{V2L}$ using the training set of our own multi-round instruction-segmentation dataset MRSeg, while keeping the weights of the CLIP image encoder and the HIPIE mask decoder frozen.

We use NVIDIA A100 GPUs for model training. We fine-tune our model with a total batch size of 16 (a per-device batch size of 2) using the AdamW optimizer (Loshchilov & Hutter, 2017) with a learning rate of $2e^{-5}$. Furthermore, we utilize stage-2 DeepSpeed accelerator (Rasley et al., 2020) and bf16 floating point precision to enhance training efficiency and reduce memory consumption.

## 5.2 EVALUATION

**Evaluation benchmarks**. For standard single-round image reasoning segmentation and detection tasks, we evaluate our model on the widely used referring segmentation and comprehension benchmarks, RefCOCO/+/g (Yu et al., 2016). We also conduct qualitative and quantitative comparisons with previous SOTA models on our multi-round referring segmentation benchmarks, based on MSCOCO (Lin et al., 2014a), PACO (Ramanathan et al., 2023) and LVIS (Gupta et al., 2019), which assess performance based on positional, interactional or hierarchical relationship queries.

**Evaluation metrics**. We use mean Intersection-Over-Union (mIoU) and cumulative Intersection-Over-Union (cIoU) as our main evaluation metrics. To assess the model's performance across multiple rounds of conversation, we track the mIoU and cIoU scores for each round's segmentation outputs.

**Baseline**. Since some baseline models, *e.g.*, LISA, do not natively support multi-round interactive segmentation, for comparisons, we adapt our multi-round validation data into their supported single-turn format by converting the $N$-turn data into $N$ single-turn instruction segmentation tasks.

## 5.3 EVALUATION RESULTS

**Mutli-round referring segmentation.** We compare the performance of SegLLM and LISA on our multi-round referring segmentation benchmarks, MR-RefCOCO/+/g. As shown in Table 1, compared to LISA (Lai et al., 2024) and GLaMM (Rasheed et al., 2024), SegLLM not only achieves 14~26% higher cIoU score across all conversation rounds but also stays stable, whereas LISA and GLaMM's performance tends to degrade in the later turns of the conversation. For example, by round 5, the performance gap between SegLLM and GLaMM widens significantly, reaching over 17.6%, 26.0%, and 14.0% on MR-RefCOCO, MR-RefCOCO+, and MR-RefCOCOg, respectively—nearly double

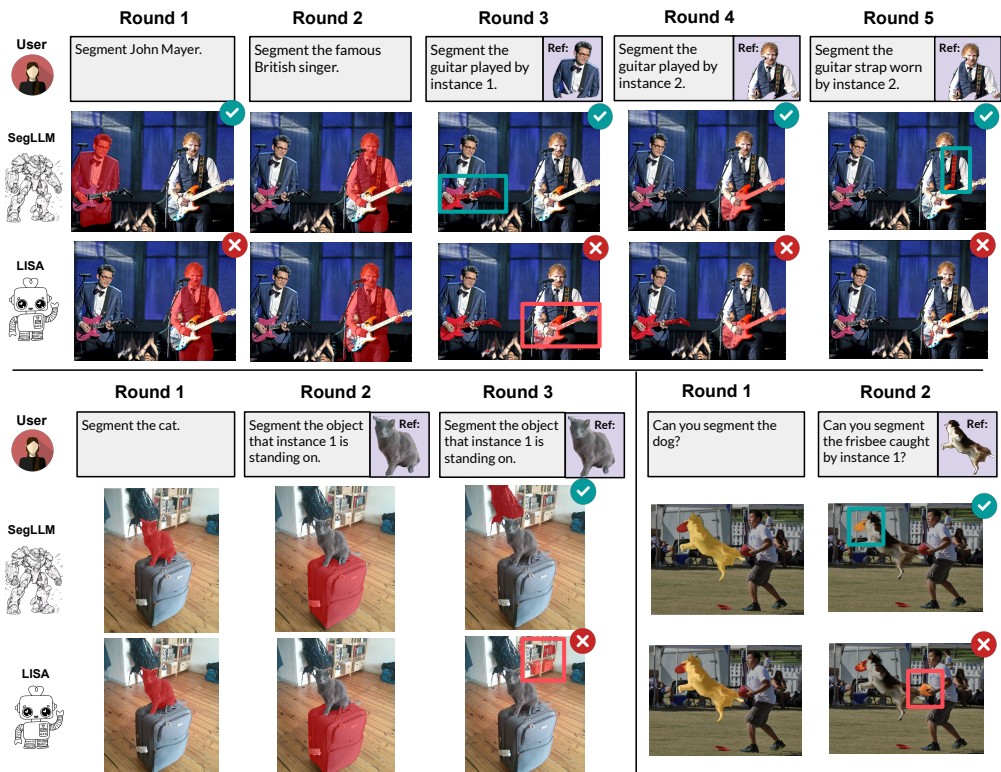

**Figure 5: Side-by-side qualitative comparison with LISA's (Lai et al., 2024) on multi-round interactive segmentation.** SegLLM not only excels in reasoning segmentation, demonstrating an understanding of world knowledge including recognition of famous individuals, as illustrated in the round 1 and round 4 results of the first demo in row one, but it also efficiently responds to questions that reference previous rounds. Ref indicates the referenced output from previous round.

| Methods | RefCOCO | | | RefCOCO+ | | | RefCOCOg | |
|---|---|---|---|---|---|---|---|---|
| | val | testA | testB | val | testA | testB | val | test |
| Shikra-13B (Chen et al., 2023) | 87.8 | 91.1 | 81.8 | 82.9 | 87.8 | 74.4 | 82.6 | 83.2 |
| VisionLLM-H (Wang et al., 2024a) | - | 86.7 | - | - | - | - | - | - |
| NExT-Chat-7B (Zhang et al., 2024) | 85.5 | 90.0 | 77.9 | 77.2 | 84.5 | 68.0 | 80.1 | 79.8 |
| **SegLLM-7B (ours)** | **90.0** | **92.1** | **86.2** | **82.2** | **85.5** | **76.1** | **83.9** | **85.9** |

**Table 3: Comparison between SegLLM and baseline models on referring expression comprehension (REC).** SegLLM not only sets a new SOTA result in referring segmentation (Table 2), but also surpasses baseline models in detection tasks, including those specifically optimized for these tasks, such as NExT-Chat-7B (Zhang et al., 2024), or models with larger LLMs like Shikra-13B (Chen et al., 2023). The evaluation metric used is the standard detection metric for REC, Acc@0.5.

| Method | Val | | Test | | Test (long query) | |
|---|---|---|---|---|---|---|
| | mIoU | cIoU | mIoU | cIoU | mIoU | cIoU |
| LISA (Lai et al., 2024) | 53.6 | 52.3 | 48.7 | 48.8 | 49.2 | 48.9 |
| SegLLM | **57.2** | **54.3** | **52.4** | 48.4 | **55.9** | **54.2** |

**Table 4:** Result comparison on the **ReasonSeg dataset**. SegLLM demonstrates superior performance, particularly on the long query subset.

the gap observed in round 1 (Table 2). For details on the evaluation protocol used to evaluate LISA on our multi-round dataset, please see Appendix B. Besides the quantitative results, Fig. 5 presents the qualitative results comparing SegLLM with LISA (Lai et al., 2024).

Why does SegLLM's **performance in later rounds sometimes exceed the first round** by 2∼4% (Table 2)? This improvement is attributed to earlier rounds helping to narrow down the search space in the image, thus enhancing the model's accuracy in subsequent queries.

**Single-round referring segmentation and expression comprehension.** As shown in Tabs. 2 to 4, SegLLM consistently exceeds previous SOTA methods, such as LISA (Lai et al., 2024), NExT-Chat (Zhang et al., 2024) and Shikra-13B (Chen et al., 2023), in the standard single-round referring segmentation and expression comprehension tasks, despite not being specifically designed for these

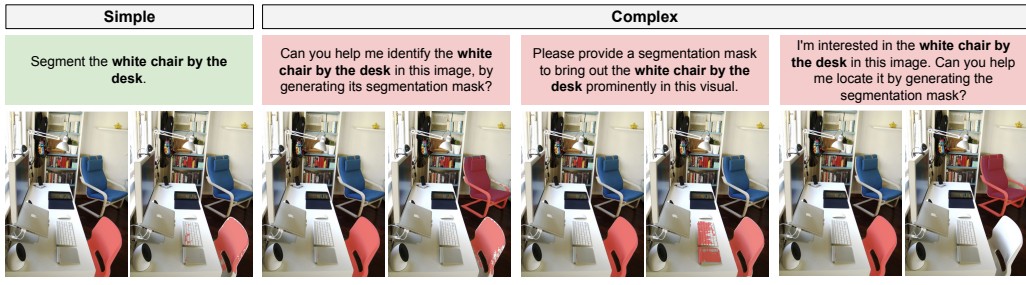

**Figure 6: Demo results that demonstrate SegLLM's robustness against varying question queries**, in contrast to LISA (Lai et al., 2024), which is sensitive to prompt phrasing. Even with simple questions presented in different templates, LISA's performance significantly declines, frequently failing to deliver correct segmentation results for most test templates. This limitation forces users to adhere to specific phrasing, such as "Segment [object descriptions]", substantially restricting the model's real-world applicability.

| Models | Multi-Round PACO (w/ LVIS) (mIoU) | | | Multi-Round PACO (w/ LVIS) (cIoU) | | |
|---|---|---|---|---|---|---|
| | LISA | SegLLM | Absolute Δ | LISA | SegLLM | Absolute Δ |
| round 1 | 34.7 | **54.9** | **+20.2** | 45.6 | **65.3** | **+19.7** |
| round 2 | 10.6 | **37.6** | **+27.0** | 15.5 | **49.7** | **+34.2** |
| round 3 | 13.7 | **32.9** | **+19.1** | 21.3 | **40.9** | **+19.6** |
| round 4 | 11.5 | **33.3** | **+21.7** | 18.7 | **39.4** | **+20.7** |
| round 5 | 11.6 | **31.6** | **+20.0** | 20.5 | **41.9** | **+21.4** |

**Table 5: Single-round referring segmentation and multi-round hierarchical image segmentation**. The Multi-Round PACO (MR-PACO) benchmark presents a significant challenge as it demands a good hierarchical understanding and the capability to precisely segment tiny masks representing parts or subparts of an object (refer to hierarchical query demos in Fig. 1). SegLLM significantly improve performance over LISA (Lai et al., 2024), demonstrating substantial improvements in both mIoU and cIoU metrics across conversation rounds.

tasks. We hypothesize that SegLLM's ability to understand the relative relationships among objects or parts within images in multi-round tasks significantly enhances its overall visual comprehension. This enhanced visual understanding capability transfers to superior performance in single-round tasks as well. However, it is worth noting that while SegLLM shows improved performance in single-round tasks, the performance gap is smaller compared to the improvements observed in multi-round tasks.

**Multi-round hierarchical segmentation** result comparison between SegLLM and LISA is conducted with our MR-PACO. As detailed in Sec. 4.1, each subsequent round may query a part or subpart of a whole object from a previous round of conversation. As shown in Table 5, compared to LISA, SegLLM obtains 10.7%∼16.2% higher mIoU and 13.2∼27.1% higher cIoU across all rounds. It is observed that the absolute model performance typically decreases in later rounds. This decline is primarily due to the progressively smaller object sizes (segment parts of an instance) in later rounds of the multi-round hierarchical segmentation task, as shown in **??**. As shown in Fig. 5, SegLLM leverages multi-round segmentation, using previous outputs to accurately identify the necktie of the person in the gray suit in round 2, as requested by the user. In contrast, LISA, lacking this contextual awareness, fails to correctly identify the person. This is further demonstrated in round 5, where SegLLM successfully segments Barack Obama's necktie from round 4, while LISA fails again.

**Robustness against question templates.** We observed that many previous studies in image reasoning segmentation, such as LISA (Lai et al., 2024) and SESAME (Wu et al., 2024), tend to overfit to the specific question templates used during training. Consequently, when these models are evaluated with diverse question templates not encountered during training, performance often significantly declines. For example, as shown in Table 6, the performance of LISA and SESAME drops by approximately 7% and 13%, respectively, when assessed using our varied templates.

To mitigate this, we intentionally diversified our question templates during the dataset generation process. As a result, our SegLLM model not only demonstrates consistent segmentation performance across diverse templates but also achieves a 5.5% higher cumulative Intersection-Over-Union (cIoU). Notably, this performance gain occurs on the single-round referring segmentation benchmark, which these prior studies were specifically optimized for. SegLLM even achieves 2.3% higher cIoU on RefCOCO+/g than (Wu et al., 2024) on its own question templates. Fig. 6 shows that LISA's performance significantly drops when asked with simple questions presented in various templates, frequently failing to produce correct segmentation results for most test templates.

| Methods | Averaged | | | Diverse | | | LISA | | | SESAME | | |
|---|---|---|---|---|---|---|---|---|---|---|---|---|
| | RC | RC+ | RCg | RC | RC+ | RCg | RC | RC+ | RCg | RC | RC+ | RCg |
| SESAME (Wu et al., 2024) | 67.4 | 57.9 | 61.4 | 66.0 | 56.9 | 60.6 | 61.4 | 51.6 | 55.6 | 74.9 | 65.1 | 67.9 |
| LISA-7B (ft) (Lai et al., 2024) | 70.1 | 61.0 | 63.0 | 67.8 | 59.0 | 62.4 | **74.7** | **64.9** | 66.1 | 67.8 | 59.2 | 60.6 |
| **SegLLM (ours)** | **79.7** | **70.0** | **72.2** | **80.2** | **70.3** | **72.6** | **80.4** | **70.7** | **72.3** | **78.6** | **69.0** | **71.6** |
| *vs. prev. SOTA* | **+9.6** | **+9.0** | **+9.1** | **+12.4** | **+11.3** | **+10.2** | **+5.7** | **+5.8** | **+6.2** | **+3.7** | **+3.9** | **+3.7** |

**Table 6: SegLLM Exhibits greater robustness to a variety of question templates in image reasoning segmentation**. Unlike previous models such as LISA (Lai et al., 2024) and SESAME (Wu et al., 2024), which tend to overfit to specific question templates encountered during training, SegLLM demonstrates improved robustness. We assess performance using the single-round RefCOCO dataset with cumulative cIoU as the evaluation metric. Notably, the templates used for evaluation were not utilized during the model training process.

| Mask-enc | Box-enc | Ref Loss | Single-Round | Multi-Round | Multi-Round (hard) |
|---|---|---|---|---|---|
| | | | RefCOCO / + / g | RefCOCO / + / g | RefCOCO / + / g |
| ✗ | ✗ | ✗ | 80.2 / 67.1 / 70.8 | 59.6 / 53.9 / 55.2 | 32.4 / 32.3 / 34.1 |
| ✓ | ✓ | ✗ | 82.3 / 72.2 / **77.8** | **75.7 / 71.3** / 68.6 | 67.6 / 67.3 / 62.8 |
| ✓ | ✓ | ✓ | **83.8 / 72.5** / 76.7 | 74.0 / 70.1 / **65.8** | **69.6 / 69.4** / 63.7 |

**Table 7: Ablation study on the effectiveness of proposed components**. Model performance evaluated on three benchmarks: (1) **Single-Round**—Referring segmentation in a single round using standard RefCOCO, RefCOCO+, and RefCOCOg datasets. (2) MRSeg: **Multi-Round**—Referring segmentation over multiple rounds, based on our custom benchmarks from RefCOCO, RefCOCO+, and RefCOCOg datasets (results show a weighted average of standard and hard subsets). (3) MRSeg (Hard): **Multi-Round (Hard)**—Focuses exclusively on the hard subset of the multi-round segmentation benchmarks. The evaluation metric used is CIoU.

## 5.4 ABLATION STUDY

**Ablation study on the effectiveness of proposed components.** We conduct an ablation study (Table 7) on our Multi-Round RefCOCO benchmark to evaluate the effectiveness of the three components we introduced in Sec. 4.2. We assess model performance across three subsets of the MR-RefCOCO dataset: 1) *Single-round*: single round referring segmentation using the standard RefCOCO/+/g datasets. 2) *MRSeg*: multi-round referring segmentation based on our MRSeg. 63) *MRSeg (Hard)*: this subset focuses exclusively on the hard subset of the MRSeg benchmarks, where understanding the reference mask is crucial for accurately segmenting the correct object. We provide more details for MRSeg (hard) in Appendix A.1.

Overall, in the MRSeg task, our proposed components lead to a significant 20% performance improvement over the baseline. In the MRSeg (hard) subset, our mask-encoding scheme achieves over 30 points higher cIoU compared to the baseline. These results highlight the effectiveness of our approach in enabling the model to interpret visual cues from user instructions and perform mask-conditioned segmentation—critical for handling complex tasks where reference masks, rather than text-based instructions, provide key information.

**The proposed components also improve results on single-round referring segmentation** as shown in Table 7. This achievement is noteworthy as the task does not explicitly require box-encoding and mask-encoding. We hypothesize that the absence of these encoding modules complicates the learning of segmentation from multi-round instructions, resulting in less stable training dynamics that affect performance even in single-round tasks.

## 6 CONCLUSIONS

We introduce SegLLM, a novel multi-round interactive reasoning segmentation model that enhances traditional segmentation models by retaining conversational memory of visual, not just textual, results. Utilizing a mask-aware multimodal large language model, SegLLM integrates previous segmentation outputs back into its input stream, allowing it to handle complex queries about relationships between objects across multiple interactions. Tested on the newly curated MRSeg, SegLLM significantly outperforms existing benchmarks in multi-round interactive segmentation by over 20% and shows a 4.7% improvement in single-round referring segmentation. These results demonstrate SegLLM's capability as a versatile model for a broad range of instruction-following segmentation tasks.

**Acknowledgment.** We thank helpful discussions with Tsung-han Wu, Xingyi Zhou, Alireza Fathi, and Cordelia Schmid. XuDong Wang and Trevor Darrell were funded by DARPA and the Berkeley AI Research (BAIR) industrial alliance programs..

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

## A APPENDIX

### A.1 DATASET DETAILS

In the section, we document further details on the dataset construction process. We also provide some statistics about our dataset.

### A.1.1 DATASET SIZE

We document the number of images sampled from each source dataset and the number of conversations generated in Table A1. Additionally, we visualize the distribution of the number of rounds for each dataset in Fig. A1.

| Datasets | Training Set | | | Validation Set | | |
|---|---|---|---|---|---|---|
| | # of Convs | # of Images | Max Rounds | # of Convs | # of Images | Max Rounds |
| RefCOCO(+/g) | 55188 | 27674 | 18 | 4263 | 2701 | 17 |
| Visual Genome | 367674 | 94221 | 2 | 40980 | 10524 | 2 |
| PACO-LVIS | 40827 | 40827 | 19 | 2178 | 2178 | 16 |
| LVIS | 71388 | 71255 | 17 | 13898 | 13898 | 18 |
| Pascal Panoptic Part | 4577 | 4577 | 17 | 4690 | 4690 | 18 |
| ADE20K | 59784 | 20196 | 1 | 5943 | 200 | 1 |
| COCO-Stuff | 340127 | 118205 | 1 | 14461 | 4999 | 1 |
| Attributes-COCO | 49036 | 36413 | 1 | 5000 | 2566 | 1 |
| ReasonSeg | 1326 | 239 | 1 | 200 | 200 | 1 |
| MRSeg (hard) | 22470 | 22470 | 1 | 1988 | 1988 | 1 |

**Table A1: Statistics of our MRSeg dataset**, including the number of overall conversations, number of images, and the maximum rounds of conversations for each dataset after processing through our dataset pipeline.

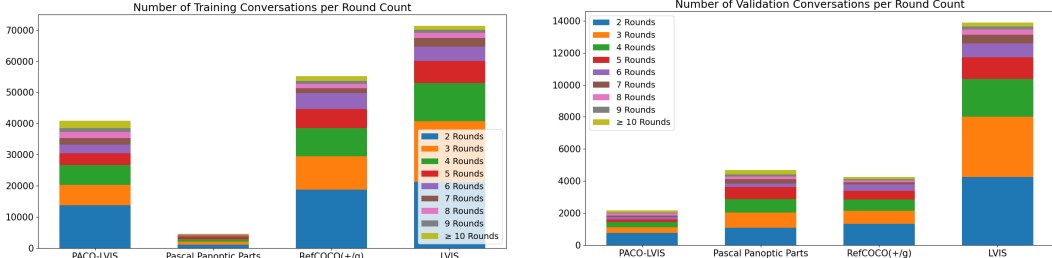

**Figure A1:** Bar-plot visualization for training and validation conversations count at different number of rounds for multi-round datasets. There are very conversations with a large number of rounds.

### A.1.2 CONVERSATION GENERATION PIPELINE

We employ different strategies to generate natural-language conversation for different source datasets. Specifically, our dataset is generated using a combination of the following methods:

- ***Hierarchical relationships based on PACO-LVIS and Pascal Panoptic Part***: In these queries, the model is asked to segment objects which are a sub-part of some output of a previous round. From each image, we randomly sample between one and four instances, and for each instance, we randomly sample between one and four parts. We initiate queries about the instance followed by questions targeting the parts of each respective instance. For Pascal Panoptic Part, we only use objects and their parts on a instance level and not a semantic segmentation level to avoid ambiguity. For both PACO-LVIS and Pascal Panoptic Part, we refer to previous round outputs with it's actual caption, e.g. ``the knife'' with probablility 50%. With the other 50% we refer to the previous round output as ``<instance i>'' or ``<the output of round i>''.

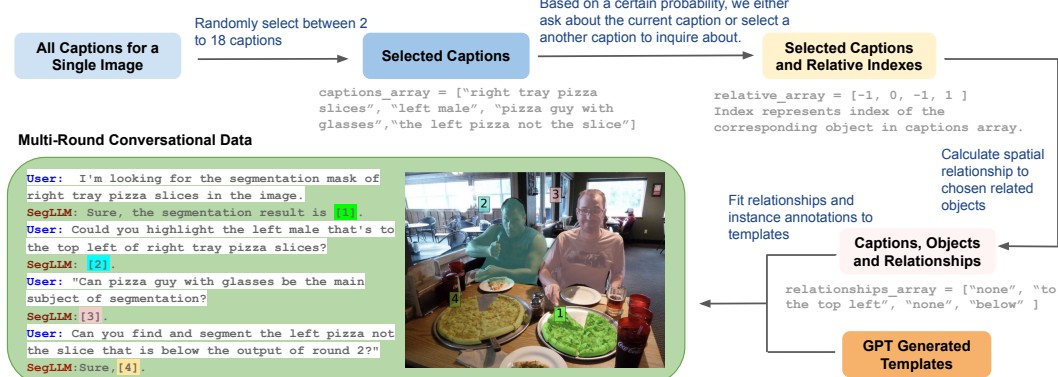

**Figure A2:** Pipeline for generating multi-round conversational data for RefCOCO(+/g) in MRSeg.

- ***Positional relationships based on Refcoco(+/g) and LVIS***: These conversations task the system with segmenting objects based on their positional relationships to the outputs from previous rounds. We randomly sample between 2 to 18 annotations per image. For each selected annotation, we either generate a query about the object itself or generate a query involving an object from previously processed instances, focusing on their relative positions calculated from their bounding box coordinates. For RefCOCO(+/g), multiple annotations may be selected for the same instance due to multiple captions available per instance. For LVIS, we select annotations where only one or two objects of that class appear in the image. When two objects of the same class are present, we detail their relative positions and add location descriptions to their captions to prevent ambiguity. We specifically choose instances not categorized under COCO classes to diversify the dataset's class variety. The probability for each round to query about an object itself is $1/3$, otherwise, we query about the current object with a reference to a previous round's output and their relative position. To assign the positional relationships, we use compare the edge and center position of the bounding boxes for the two instance we are trying to assign a relationship to.

There are 9 total possible positions two instances can have (the same as, overlapping with, to the left/right, above/below, to the top/bottom left/right of). Similar to Hierarchical Queries, we refer to previous round outputs with it's actual caption, e.g. ``the woman on the left'' with probablility 50%. With the other 50% we refer to the previous round output as ``<instance i>'' or ``<the output of round i>''. A detailed pipeline for how RefCOCO(+/g) dataset is sampled can be see in  Fig. A2

- *MR Seg(hard)*: For each RefCOCO image, we identify cases where there are two instances of the same class within the image. From these, we select a pair of instances and construct two single-round conversations. Given two instances, X and Y, of the same class in the image, we create the following conversations:

  - Conv 1:  `[IMAGE] [ENCODE X] Please segment the other <class name>` → `Sure, [DECODE Y]`
  - Conv 2:  `[IMAGE] [ENCODE Y] Please segment the other <class name>` → `Sure, [DECODE X]`

  We have 10 different templates for the training and 5 templates validation/test for MR Seg(hard).

- *Interactional relationships based on Visual Genome*: We adopt Visual Genome (VG), utilizing its relationship annotations to construct conversations that emphasize interactional dynamics rather than merely positional relationships. We sample up to four relationships per image. Each relationship prompts a two-round conversation: the first round involves segmenting the subject, and the second round involves segmenting an object based on its relationship to the subject. Since VG also only provides bounding box labels, we generate masks for selected instances using SAM.

### A.1.3   DETAILS OF GPT4 USAGE

We prompt GPT-4 models for generating captions for attribute-based descriptions as well as for cleaning grammar errors in our dataset. The detailed instructions and specific model we used can be found in  Table A2 and  Table A3. For the attribute-based description, we crop COCO images to only contain the specified instance, feeding the cropped image and it's class name to GPT to generate a description. For language correction, we found that grammar correction is often erroneous but can be a lot of accurate if we go through the data twice to double check.

### A.1.4   MORE DISCUSSIONS ON DEMO OUTPUTS

In Fig. A3, example **A** illustrate the necessity of our Mask-Encoding Scheme, to avoid the ambiguity that may arise in cases where multiple instances of the same class are present in the image. Round 2 and round 3 in example **A** show that without our mask encoding mechanism to supply information about the person segmented from round 1, since there are multiple laptops and chairs present in the image, confusion arises as to which specific laptop or chair the user is referring to in the query prompt. Therefore, without the guiding information from the mask encoding, LISA seems to naively guess the incorrect laptop in round 2, and does not generate a comprehensible segmentation mask in round 3. In contrast, the mask encoding guides our model to correctly segment the requested objects. Similarly, in round 4 and round 6, our model was able to successfully segment the keyboard of the laptop from round 3 and the person setting on the chair from round 5.

This phenomenon is again demonstrated in **B** in Fig. A3. Since there are two women, both carrying bags and holding an umbrella in the image, our Mask-Encoding Scheme again resolves this the ambiguity and allows the user to conveniently specify the bag and the umbrella requested in round 2 and round 3 are carried and held by the person from round 1. As before, the awareness of previous round outputs enables our model to segment the correct objects, whereas LISA guesses the incorrect objects due to the lack of this awareness.

Example **C** demonstrates that our model is not limited to multi-round prompting, and can produce accurate segmentation results via direct, single-round prompts as well. In the indirect case, we first ask the model to segment the dog during the first round of the conversation. Then, in the second round, we ask a follow up question to guide the model to segment the Frisbee that is caught by the dog from round 1. However, tin the direct case, we straight away ask for the Frisbee that is caught by the dog. In comparison, our model succeeds in both the direct and indirect case, whereas LISA fails to segment the correct Frisbee instance in either cases. This shows that our multi-round comprehension capability is not a limitation but an addition.

```
payload = {
    "model": "gpt-4-turbo-2024-04-09",
    "messages": [
      {
        "role": "user",
        "content": [
          {
            "type": "text",
            "text": f"Can you focus on describing the {class_name} in
                the image? Can you format your output in a two item
                array, such that the first index is an abstract
                description without any class name, such as 'has a pizza
                sitting on top of it' or 'is wearing a beige t-shirt'
                and the second index is the exact classname for the
                object, such as 'a dining table' or 'a man'."
          },
          {
            "type": "image_url",
            "image_url": {
              "url": f"data:image/jpeg;base64,{base64_image}",
              "detail": "low"
            }
          }
        ]
      }
    ],
    "max_tokens": 200
  }
```

**Table A2:** Our full prompt to the GPT-4-turbo-2024-04-09 model for generating abstract descriptions

Lastly, we note that round 3 and round 6 of example **A**, round 2 and round 3 of example **B** and round 2 of example **C** demonstrate our model's understanding of *interactional relationships* as introduced in Sec. 4.1 and round 4 demonstrates the *hierarchical relationship* introduced in Sec. 4.1.

## B DETAILS OF COMPARISON WITH LISA

Since Lisa does not naively support multi-round training, to ensure fairness, we employed two different approaches:

- Approach One: We substitute the mask and bounding box encoding tokens of the reference instance with the word "mask". For example, a query in MR-RefCOCO dataset "Segment the person to the left of `<mask> <box>`." would be converted to "Segment the person left to the mask."

- Approach Two: We substitute the mask and bounding box encoding tokens with the description of the reference instance. For example, a query in MR-RefCOCO dataset "Segment the person to the left of `<mask> <box>`." would be converted to "Segment the person left to the dog chasing after a butterfly." (where `<mask> <box>` are encoding tokens of the reference instance "the dog chasing after a butterfly")

We report results on MR-RefCOCO in Table A4. SegLLM outperforms both alternative approaches 1 and 2. Furthermore, we find that LISA performs worse using approach 2 compared to approach 1, despite the inclusion of the description of the reference instance. We suspect that this may be due to LISA being trained on data that focuses on 1 instance, hence the presence of description for two instances, the target and the reference instance, may cause more confusion than guidance. Regardless, in our main table Table 1, we report LISA's performance on our MR-RefCOCO/+/g benchmark using the best approach for LISA, approach 2.

```
Round 1:
response = client.chat.completions.create(
          model="gpt-4o-2024-05-13",
          response_format={ "type": "json_object" },
          messages=[
              {"role": "system", "content": "You are a helpful
                  assistant designed to output JSON."},
              {"role": "user", "content": f"Can you fix any errors and
                  make the sentence sound like natural English, and
                  provide our output in a dictionary of format
                  'corrected'=CORRECT_SENTENCE? here is the sentence I
                  want you to correct, '{sent}'"}
          ]
      )
Round 2:
    response = client.chat.completions.create(
          model="gpt-4o-2024-05-13",
          response_format={ "type": "json_object" },
          messages=[
              {"role": "system", "content": "You are a helpful
                  assistant designed to output JSON"},
              {"role": "user", "content": f"Here is the original
                  sentence: '{sent}'. Here is the corrected sentence:
                  '{corrected_sent}'. Does the corrected sentence have
                  the same meaning as the original? If yes, please
                  output ['Same', 'None']. If no, please output
                  ['Different',
                  '<corrected_with_same_meaning_as_original>']."}
          ]
      )
```

**Table A3:** Out full prompt to the gpt-4o-2024-05-13 model for grammar correction. We use a two-round approach, feeding GPT's first round answer back to itself to be self-corrected.

**Table A4:** Comparison of SegLLM, Lisa(Approach 1), and Lisa(Approach 2) on MR-RefCOCO evaluation set.

|         | SegLLM | Approach 1 | Approach 2 |
|---------|--------|------------|------------|
| round 2 | 81.9   | 60.6       | 55.9       |
| round 3 | 81.7   | 58.9       | 54.7       |
| round 4 | 78.4   | 61.3       | 56.7       |
| round 5 | 80.3   | 61.0       | 57.8       |
| round 6 | 74.5   | 60.7       | 57.7       |
| round 7 | 69.3   | 54.4       | 45.6       |
| round 8 | 70.5   | 51.9       | 50.3       |

## C  LICENSE

We makes use the following models: CLIP (MIT license), LLAMA 2 (Llama 2 Community License Agreement), Vicuna (Apache2 license). BLIP-2 ( BSD-3-Clause license)

We use the following dataset COCO (Attribution-NonCommercial-ShareAlike 4.0 Internationa), RefCOCO (Apache-2.0 license), Visual Genome (Creative Commons Attribution 4.0 International License.), PACO (MIT License), Pascal-Panoptic-Parts ( Apache-2.0 license), LIVIS (CC BY 4.0 + COCO license).

## D  LIMITATIONS

Although we have shown some promising quantitative results in the novel multi-round reasoning segmentation task, our method exhibits several limitations upon qualitative examination, as shown in Fig. A4 and discussed in detail below. In addition to revealing potential weaknesses within

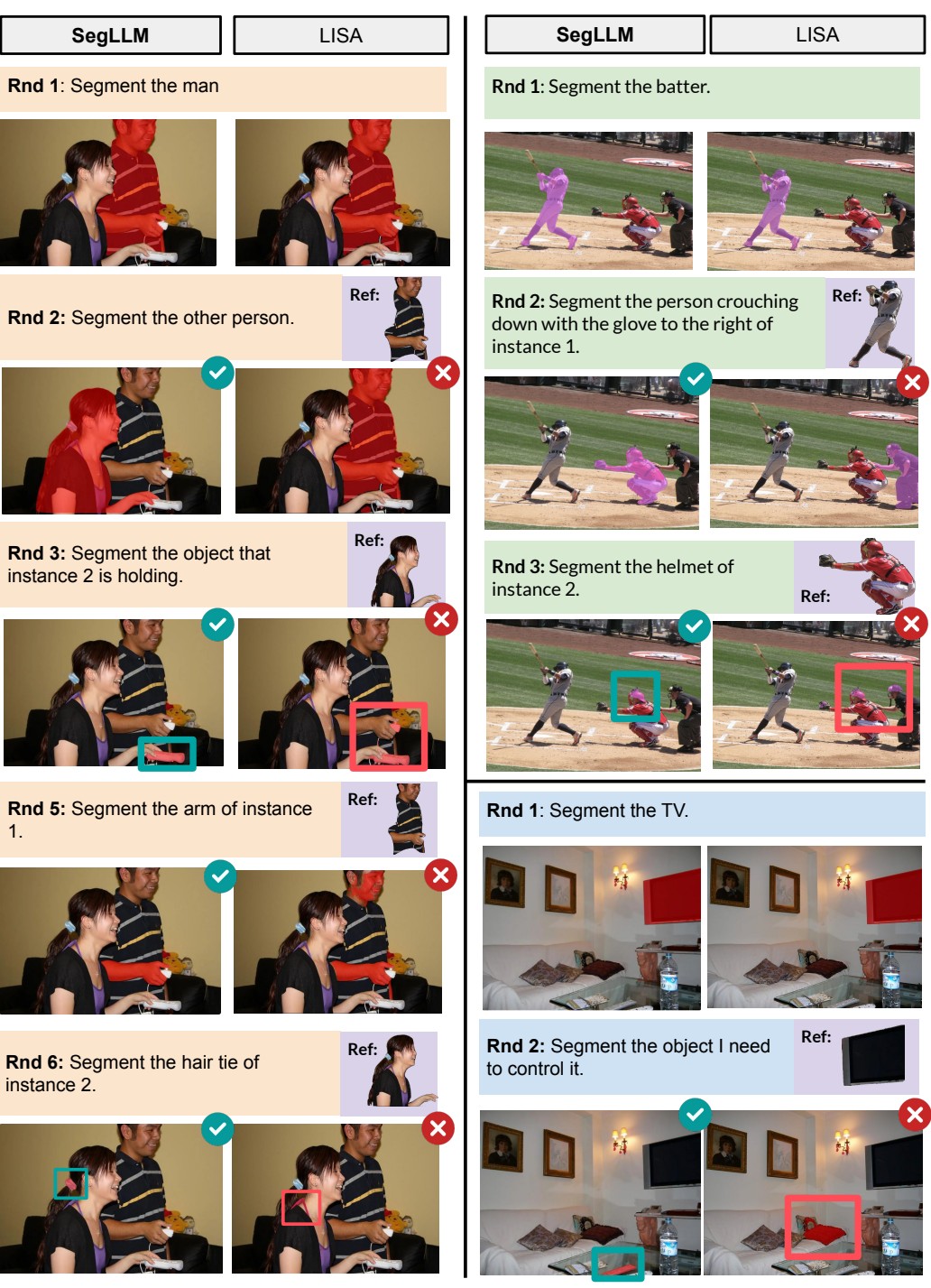

**Figure A3:** Additional side-by-side comparison with LISA. This shows that without awareness of segmentation outputs from previous rounds, LISA struggles to identify the correct instance requested by the user, when there is ambiguity. `Ref` indicates the referenced output from previous round.

our proposed model components, perhaps the existence of failure cases in qualitative evaluation despite the impressive performance in quantitative evaluation indicates that our dataset construction methodology also requires improvement such as including harder test samples. It is our hope that these findings will encourage further research along the direction of multi-round segmentation,

aiming to improve the incorporation of conversation and segmentation history, address some of these limitations and extend upon our initial proposals and approaches.

**Sensitivity to conversation history order.** Given fixed input queries for subsequent rounds, when the order of the previous rounds in the conversation history are permuted, the model's output is not consistent. As shown by conversation 1 in Fig. A4, in rounds 1 and 2, version A first asks for the left paddle board and then the right paddle board, whereas version B first asks for the right paddle board and then the left paddle board. However, in rounds 3 and 4, both version have the same queries, first asking for the person standing on top of the paddle board from round 1 followed by the person standing on top of the paddle board from round 2. Despite having the same input queries for rounds 3 and 4, the model only succeeds in segmenting the correct instance in version A and fails to do so in version B. From the user's standpoint, the two versions are equivalent, since the user is equally likely to start the conversation by asking for the left paddle board, as for the right paddle board. However, the model's behavior is not invariant under the permutation of the conversation history, suggesting that the robustness of our model can be improved.

**Sensitivity to input query order.** Symmetric to the previous case, given a fixed conversation history, if the order of the queries in subsequent rounds are permuted, then the model's output is also no consistent. As shown by conversation 2 in Fig. A4, in rounds 1 and 2, both version A and version B first asks for the person wearing red followed by the person wearing blue. Then, in rounds 3 and 4, version A first asks for the chair that the person from round 1 is sitting on (query 1), followed by the chair that the person from round 2 is sitting on (query 2), and version B asks the same queries but in the opposite order. However, despite the queries being the exact same, simply switching the order of these two queries causes the model to only succeed in segmenting the correct instance in version A and fails to do so in version B. Again, from the user's perspective, the two versions are equivalent, since the user is equally likely to first request an instance related to the output from round 1, as to first request an instance related the output from round 2. However, the model's behavior is not invariant under the permutation of the input queries in subsequent rounds, suggesting that the robustness of our model can be improved.

**Independence of encoding information.** Although we quantitatively showed that using our proposed mask encoding component to re-introduce past segmentation outputs into the model's input stream can surpass the performance of other models without this component on our multi-round segmentation benchmarks in Sec. 5.3, as well as verified its effectiveness through conducting ablation study in Sec. 5.4, qualitative evaluations show that the mask encoding information provided by this component may be underutilized by the model. As shown by conversation 3 in Fig. A4, in rounds 3 and 4, the helmet belonging to the kid from round 1 and round 2 are asked, respectively. In version A, the mask encodings corresponding to the reference instance (the kid from round 1 or round 2) are supplied to the model, whereas in version B, the relevant encoding information were not provided. However, despite this lack of encoding information, the model is able to successfully segment the correct instance in version B as well as in version A. This suggest that perhaps the textual information of "instance 1" is sufficient for the model to reason that the helmet requested in round 3 corresponds to the one belonging to kid from round 1. Alternatively, the model may just be randomly guessing, which in this example has a success probability of 0.5. Regardless, it may be worthwhile to re-think the design of our multi-round dataset. Perhaps increasing the level of ambiguity by filtering for images where $> 2$ instances of the same class are present can force the model to rely on mask encoding information to reason about the requested instance.

**Sensitivity to positional keywords.** Lastly, during qualitative evaluation, we also observed that our model can be highly sensitive and reliant on positional keywords, such as "left" or "right". As shown in single round conversations 1A and 1B in Fig. A4, the input queries are purposefully constructed to include the positional word "left" whilst requesting for the person on the right (1A), and including word "right" whilst asking for the person on the left, respectively. Indeed, the model falls for this trick, segmenting the person on the left and on the right, whenever the word "left" or "right" is present in the prompt, respectively, failing to comprehend the referring expression. This indicates that perhaps the model is paying the most attention to such positional keywords, instead of understanding the entire expression. Moreover, as shown in single round conversations 2A and 2B in Fig. A4, the model is able to correctly segment the request instance when given a referring expression that includes a positional keywords "right" (2A), but fails to do so when given an alternative referring expression that doesn't include any positional keywords. This further indicates that currently our model relies heavily on positional keywords that appear in the referring expression for the request instance, thus

limiting its accuracy and generalization beyond expression that do not contain such keywords. As mentioned previously, perhaps re-designing the construction of our dataset, such as by reducing the number of samples involving Positional Relationships (see Sec. 4.1) and introducing more complex relationships between instances may alleviate this dependency on positional keywords.

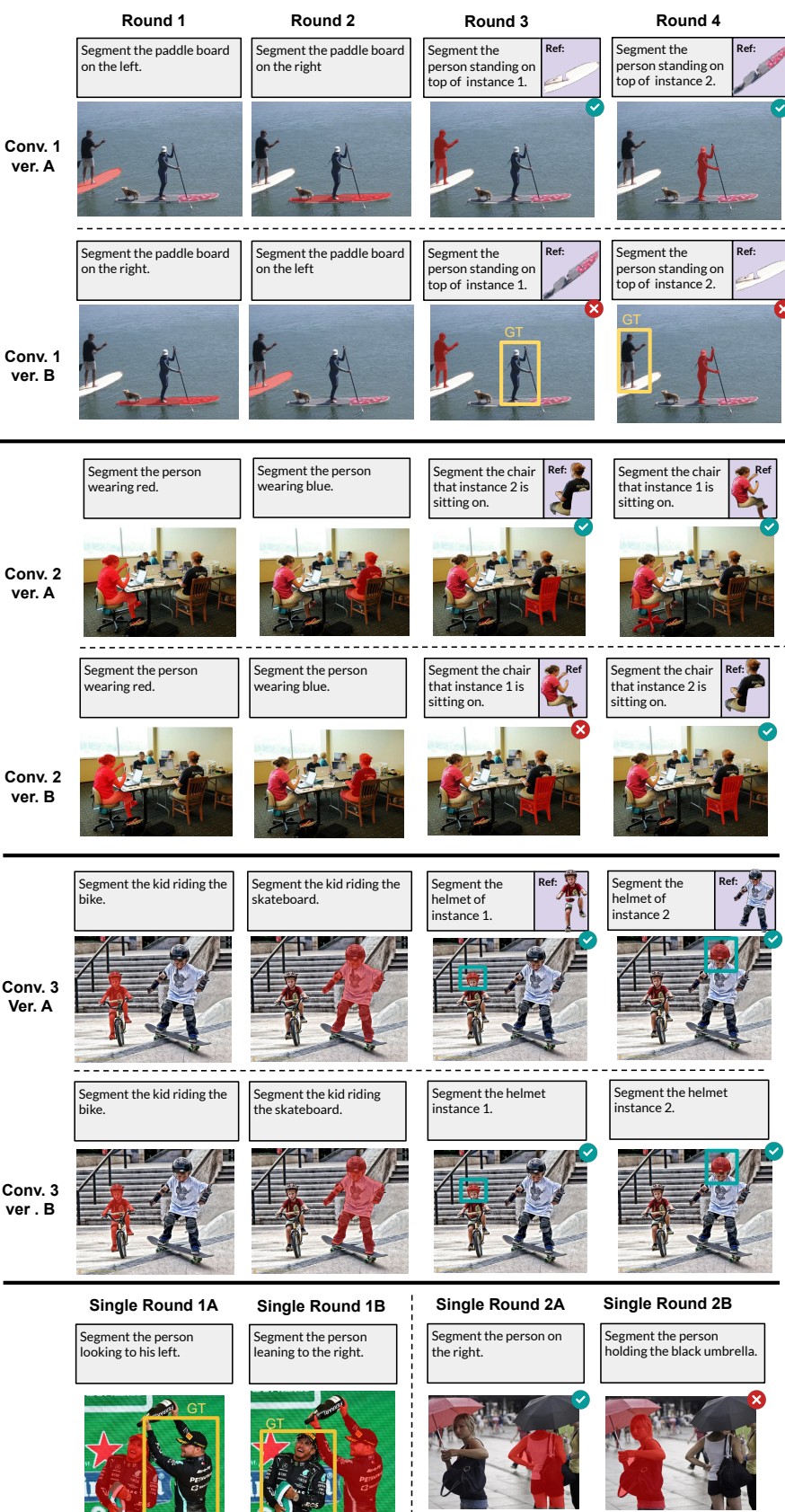

**Figure A4: Limitations exhibited by model during qualitative evaluation.** (Top row) Sensitivity to order of conversation history, given fixed queries in subsequent round. (2nd row) Sensitivity to order of queries in subsequent rounds, given fixed conversation history. (3rd row) Independence of mask encoding information. (Bottom row) Reliance on positional keywords in referring expression of query.

