# OpenReview forum: "SegLLM: Multi-round Reasoning Segmentation with Large Language Models"
_ICLR.cc/2025/Conference — ICLR 2025 Poster_

### Official Review · Reviewer_g3qz · 2024-11-02

**Soundness:** 3
**Presentation:** 3
**Contribution:** 3
**Rating:** 6
**Confidence:** 5

**Summary:**

This paper introduces SegLLM, an advanced interactive reasoning segmentation model that leverages **multi-round dialogue capabilities** in Large Language Models. SegLLM outperforms existing models by enhancing interactional reasoning and memory retention, demonstrated on a new benchmark, MRSeg, with substantial gains in accuracy metrics.

**Strengths:**

1. This paper proposes a novel integration of mask-aware LLMs for segmentation, enabling iterative, memory-based multi-round interactions.

2. SegLLM outperforms state-of-the-art segmentation models like LISA with significant performance margins across single and multi-round tasks.

3. Paper provides a comprehensive MRSeg dataset, structured for interactional and hierarchical segmentation tasks.

4. Paper demonstrates robustness against diverse question templates, improving the practical applicability of multi-modal conversational AI.

**Weaknesses:**

1. Using vision prompts for referring segmentation is a good approach. However, I would like to see a more comprehensive ablation study in Table 7 of the main paper, exploring the effects of different types of vision prompts, such as using only the mask encoder or only the bounding box encoder.

2. It would be valuable to understand the performance impact of directly using the `[SEG]` token from the previous round as the visual prompt for the current round. This approach might reduce the computational overhead introduced by the mask and bounding box encoders.

Minor Comment: Typically, table captions are placed at the top rather than the bottom of the table.

**Questions:**

See weaknesses.

---

### Official Review · Reviewer_PsJ9 · 2024-11-02

**Soundness:** 3
**Presentation:** 4
**Contribution:** 3
**Rating:** 6
**Confidence:** 4

**Summary:**

This paper introduces SegLLM, an advanced multi-round interactive segmentation model that improves LLM-based segmentation by incorporating conversational memory of past visual and textual outputs. Utilizing a mask-aware multimodal LLM, SegLLM refines its inputs with previous segmentation results, allowing it to handle complex queries about object relationships across multiple interactions. Testing shows SegLLM outperforms existing methods in both interactive and single-round segmentation tasks.

**Strengths:**

1. This paper proposes a multi-round reasoning segmentation task, which extends the boundaries of traditional reasoning segmentation and introduces new avenues for exploration within the community.
2. The paper presents the SegLLM model, which supports both bounding box and mask inputs, expanding the capabilities of large segmentation models. For the multi-round reasoning segmentation task, the authors carefully designed special tokens and a corresponding mask decoder. These modifications, in my view, represent an extension of the LISA paradigm with practical significance.
3. The paper constructs a multi-round reasoning segmentation dataset, drawing from diverse sources and following a well-designed pipeline, thus advancing research in this area through a robust data contribution.
4. The experiments are thorough, showcasing the task’s difficulty while effectively validating the proposed modules.

**Weaknesses:**

1. Methods like LISA employ LoRA for fine-tuning; does this paper adopt a similar approach? If not, could the observed performance improvement over LISA be attributed to a more advanced fine-tuning strategy?
2. The paper lacks an ablation study with masks and bounding boxes separately as inputs, and it is unclear whether using points or drawing trajectories as prompts would be viable.
3. There are minor errors in the text, such as on lines 427–428 and line 470.

**Questions:**

Will the code, dataset, and details on data processing be made open-source?

---

> ### Comment · Reviewer_PsJ9 · 2024-12-03
>
> Thanks for the detailed responses. My concerns have been largely addressed, and the authors' contributions have been clearly articulated. Therefore, I will maintain my previous score of 6.

---

### Official Review · Reviewer_rtU2 · 2024-11-02

**Soundness:** 3
**Presentation:** 4
**Contribution:** 3
**Rating:** 6
**Confidence:** 4

**Summary:**

This paper proposes SegLLM a new Multimodal Language Model capable of performing multi-turn segmentation tasks based on language inputs. The paper introduces two new mechanisms to ease the multi-turn segmentation by allowing the LLM to get mask and bounding box tokens as inputs. This mechanisms allows the LLM to understand multi-turn instructions that can refer to previously predicted masks. Along with the method, the paper is paired with a new dataset to train multi-turn segmentation tasks. The dataset was created automatically by using current single-turn datasets and leveraging the capabilities of Language Models to adjust input prompts.
The results show that SegLLM is capable of solving multi-turn segmentation better than other alternatives, while also improving at single-turn segmentation benchmarks.

**Strengths:**

The paper is really well presented and written. The ideas are clear and the problem that the paper is trying to solve is clear from the get go.
The method is also clear and simple to understand, while still being technically sound. The validation of the method using the self-created dataset and benchmark is also a great contribution for the community. The ablation studies are solid and conclusive and validate the design choices.

**Weaknesses:**

There are a few of the paper that were not 100% clear to me and I would appreciate some clarifications.
1. The metric for the multi-turn segmentation is not clear to me. Is the goal of multiple turn to simply segment correctly the object that is referred in the last turn? Or does the model have to get every single middle step correctly as well? Additionally, is every turn referring to only one instance at a time?

2. Is is not clear how were the baselines evaluated on the proposed MR-RefCOCO dataset. I did not see the details on whether the baselines were trained using the same training split as SegLLM or not. If the training split was not used to further refine these method then some conclusions become less strong. For instance, the fact that SegLLM is more robust to query diversity is mostly due to the dataset but not the method itself, then just training LISA on this dataset would equip it with similar robustness to query diversity. Additionally, that would explain the big gap between the previous methods and the proposed one. This needs a more in-detail explanation to understand the results of the paper in detail.

3.  The ablation study although looks good in terms of numbers, is not clear in terms of excecution. It is not clear to me how can SegLLM perform multi-turn segmentation without having the module that allows it to "see" the previously segmented masks.  How would the model be able to understand queries that refer to previously segmented masks? This whole table needs more details on how would the model operate without some of these critical components.

4. Related to the previous question, it is also not clear how other method can perform the task without the mechanisms to refer to previously segmented masks. Do you feed an image with the painted instance back to the model? How is it performed?




Additional notes:

When showing the queries to ChatGPT API in the supplementary materials, I would suggest to simply put the question/template and answers instead of the whole command used. It is not needed and it makes it really hard to read.

**Questions:**

There are a few aspects of the paper that were not entirely clear to me, and I would appreciate some clarifications:

1. **Metric for Multi-turn Segmentation**:
   - Is the goal of multi-turn segmentation to segment only the object referred to in the final turn, or is the model expected to get each intermediate step correct as well?
   - Additionally, does each turn in the multi-turn setup refer to only one instance at a time?

2. **Evaluation of Baselines on MR-RefCOCO Dataset**:
   - How were the baselines evaluated on the proposed MR-RefCOCO dataset? I did not see details on whether the baselines were trained using the same training split as SegLLM.
   - If the training split was not used to refine these methods, could this weaken the strength of some conclusions? For instance, if SegLLM’s robustness to query diversity is largely due to the dataset rather than the model itself, wouldn't training LISA on this dataset also equip it with similar robustness?
   - This additional context would also clarify the performance gap observed between previous methods and the proposed one.

3. **Ablation Study Execution**:
   - Although the ablation study results look good numerically, I find some of the details around execution unclear. For example, how does SegLLM perform multi-turn segmentation without the module that enables it to "see" previously segmented masks?
   - Without access to these masks, how would the model handle queries that refer back to prior segmentations? More details on how the model operates without these critical components would make this section much clearer.

4. **Multi-turn Segmentation by Other Methods**:
   - Related to the previous question, how do other methods in this study perform the task without mechanisms to refer to previously segmented masks?
   - Do you feed an image back into the model with the segmented instance painted in? If so, could you provide more details on this approach?


I would love to hear these clarifications from the authors. I believe the paper is already very good, and making it crystal clear for the reader would make it a great work.

---

### Meta-Review · Area_Chair_H7Wu · 2024-12-19

**Metareview:**

The paper proposes a new multimodal language model for interactive multi-turn segmentation exploiting conversational memory.
The paper clearly presents the idea, which is well motivated, and demonstrates the effectiveness of the method through a comprehensive set of experiments. The reviewers initially requested to clarify several points and to perform some additional ablation studies, which the authors properly responded to. The paper initially received all positive reviews and maintained the scores throughout the discussion period. Therefore, the AC also recommends the acceptance of the paper.

**Additional Comments On Reviewer Discussion:**

The author rebuttals resolved most concerns the reviewers raised. In consequence, all the reviewers maintained the original positive ratings (6) after the discussions. Two common points raised by the reviewers but resolved by the authors are some clarifications about the method and requests for more comprehensive ablation studies.

---

### Decision · Program_Chairs · 2025-01-22

Accept (Poster)